# Sterile Fecal Microbiota Transplantation Boosts Anti-Inflammatory T-Cell Response in Ulcerative Colitis Patients

**DOI:** 10.3390/ijms25031886

**Published:** 2024-02-04

**Authors:** Anton Chechushkov, Pavel Desyukevich, Timir Yakovlev, Lina Al Allaf, Evgeniya Shrainer, Vitalyi Morozov, Nina Tikunova

**Affiliations:** 1Federal State Public Scientific Institution “Institute of Chemical Biology and Fundamental Medicine”, Siberian Branch of the Russian Academy of Sciences, 630090 Novosibirsk, Russiashrayner_ev@cnmt.ru (E.S.); doctor.morozov@mail.ru (V.M.); 2Advanced Engineering School, Novosibirsk State University, 630090 Novosibirsk, Russia; 3Autonomous Non-Commercial Organization “Center of New Medical Technologies in Akademgorodok”, 630090 Novosibirsk, Russia; 4Department of Natural Sciences, Novosibirsk State University, 630090 Novosibirsk, Russia

**Keywords:** ulcerative colitis, sterile fecal microbiota transplantation, regulatory T cells, immune response

## Abstract

Ulcerative colitis is a chronic immune-mediated disease of unclear etiology, affecting people of different ages and significantly reducing the quality of life. Modern methods of therapy are mainly represented by anti-inflammatory drugs and are not aimed at a specific pathogenetic factor. In this study, we investigated the effect of transplantation of sterile stool filtrate from healthy donors on the induction of anti-inflammatory immune mechanisms. It was shown that performing such a procedure in patients with ulcerative colitis caused the appearance of T helper cells in the blood, which reacted to the content of sterile stool filtrates in an antigen-specific manner and produced IL-10. At the same time, cells of the same patients before therapy in response to the addition of sterile stool filtrates were less reactive and predominantly produced IL-4, indicating its pro-inflammatory skewing. The obtained data demonstrated the effect of an anti-inflammatory shift in the T-helper response after transplantation of sterile stool filtrate, which increased and persisted for at least three months after the procedure.

## 1. Introduction

Ulcerative colitis (UC) is a multifactorial disease primarily driven by immune system hyperactivation. However, neither the particular pathogen nor the antigen set has been described as a strong driving factor. The pathogenesis of UC is mainly immune-related inflammation, caused by hypersensitivity reactions in the sub-epithelial layers of the colon. UC, in contrast to Crohn’s disease, is primarily driven by Th2 type immune responses, making it more similar to hypersensitivity type 4 responses than the typical bacteria-specific responses that are primarily caused by Th17 type cells. This immune reaction is characterized by the hyperactivity of CD4^+^ T-cells, specifically recognizing particular antigens, not necessarily of a microbial nature. However, the Th2-type response is also a hallmark of any type of long-lasting chronic inflammation, including both bacteria-specific and autoummune-type responses.

To date, the only available therapy is a spectrum of anti-inflammatory medications, ranging from moderately light non-steroid anti-inflammatory drugs to severe hormonal and cytostatic drugs. Despite the progress made with the introduction of novel anti-inflammatory immunotherapy, the overall remission net rate remains low [1]. One of the potential explanations is that the state of immune system activity is not the primer for the inflammation in this case [2], and blocking one or the other immune pathway does not play a long-term role, as another type of response may be activated towards an unknown factor or a group of factors. In any case, it has been shown that inflammatory bowel diseases (IDB) is associated with a relative deficiency of regulatory T-cell responses [3,4], making them an autoimmune spectrum disease that is characterized by a lack of immune suppression towards self-antigens. However, no prognostic or actionable autoantigen has been proposed yet as a strong clinically significant biomarker.

A novel view of the problem implies that microbiota alteration may be a driver of the observed alterations. The intestinal microbiota itself serves as a relevant immune-training compartment [5], providing both innate (PAMPs, MAMPs) and specific (bacteria-derived) antigens [6]. The pattern of microbiota alteration characterized by the increase in relative abundance of the Firmicutes phylum compared to the Bacteroidota phylum represents the typical finding during clinical examination of a patient’s gut microbiome using 16S ribosomal RNA [7]. Also, a direct cause-and-effect relationship has been shown for Bacteroidotes-mediated Treg upregulation [8], while bacteria of the Firmicutes phylum are recognized to be associated with autoimmune spectrum diseases such as rheumatoid arthritis [9,10]. However, other studies do not support the Firmucited/Bacteroidotes ratio as a significant marker of the particular inflammatory bowel disease (IBD) [11].

As a method of treating patients with IBD, Fecal Microbiota Transplantation (FMT) has been suggested. The concept was initially based on the idea that the simple transfer of bacterial communities from healthy donors to patients with IBD may shift the inflammatory state because of the induction of the right bacteria, which brings immune-suppressive capacity. Nevertheless, to date, no clinical trial has shown unambiguous outcomes for that type of therapy in patients with UC and Crohn’s disease [12]. It should be noted that microbiota transplantation, being the method under investigation, is not applicable to patients with a severe inflammatory state, which limits the data available on the potential treatment outcomes. The major source of negative outcomes of FMT may be the fact that the microbiota compositions may differ drastically [13], and the patient-specific metabolic profiles are not appropriate for the donor microbiota community “engraftment” [14]. Overall, the FMT was prone to be more successful in patients with non-severe onset or prolongation of remission duration in chronic UC patients [11].

One of the promising possibilities is that bacterial transfer itself is not as important for FMT to be efficient as the metabolite profile or bacteriophage content [15,16,17]. In contrast to the idea that non-sterile FMT may contain bacteria that directly influence the immune response, the idea of bacteria-free FMT is based on its regulatory rather than bacterial-engraftment effect. Thus, FMT has been shown to alter the ratio of helper T cells in favor of regulatory T cells and other IL-10-producing T cells in a murine model of dextran sulfate-induced colitis [18]. Applied to humans, FMT in patients with inflammatory bowel diseases resulted in increased IL-10 production and a decrease in IL-17 blood level [19]. Some researchers attribute these effects to the colonization of healthy donor intestinal microbiota, accompanied by increased production of a number of metabolites that promote the formation of regulatory T cells. At the same time, other studies have shown that intestinal colonization in FMT is not a key factor in the beneficial effects of therapy [12]. 

The aim of this study was to investigate the potential of sterile stool samples containing no bacterial cells to influence anti-inflammatory helper T cell activation in patients after an experimental Sterile Fecal Microbiota Transplantation (SFMT).

## 2. Results

### 2.1. Patients

This study included eight adult patients (Table 1) who agreed to undergo sterile intestinal microbiota transplantation and met the inclusion criteria for patients in the study design. Their age varied from 20 to 51 years; they had confirmed diagnoses of moderate and severe-moderate UC, and the duration of chronic UC was more than one year (Table 1). The diagnosis was confirmed on the basis of the fecal calprotectin level analysis, fibrocolonoscopy (with anesthesia) data, and histological examination of biopsy specimens taken from different parts of the colon and ileum. The severity of UC was determined on the basis of the Mayo scale, taking into account the frequency of defecation, the presence of blood in the stool, the state of the intestinal mucosa (determined endoscopically), and the general assessment by a gastroenterologist. When the score on the Mayo scale was 2–5 or 6, the UC was considered moderate or severe-moderate, respectively. In accordance with the patient’s roadmap, each patient underwent a general blood test, biochemical analysis (total protein, urea, and bilirubin content), and determination of the concentration of C-reactive protein (Table 1).

Patients with severe UC were excluded from this study in order to unify the experimental group. In addition, severe UC is typically provided with severe immune suppressive therapy, which may inadequately interfere with this study’s aims. Moreover, patients with severe UC are at a potentially greater risk of exacerbating inflammation due to an increased immune system response.

### 2.2. Immune Status of Biological Samples and Experimental Settings 

The hypothesis of this study suggests that the administration of sterile stool filtrates can cause an immunomodulatory effect due to antigenic and adjuvant factors in the microbiota of healthy donors. As the evaluation of bowel T-cells is a challenging task (totally not applicable in patients with mild disease), we assessed the memory and activation state of peripheral T-cells derived from peripheral blood mononuclear cells stimulated with the sterile content of donor stool samples and recipient stool samples obtained at different time points after the treatment. CD4^+^ lymphocytes and the concentration of the cytokines IL-4 and IL-10, which play a role in maintaining or switching off the inflammatory response, were studied. Biomaterial from patients was collected at three time points: before transplantation, 1 and 3 months after the procedure (Figure 1a).

In order to establish these effects, peripheral mononuclear cells obtained at different experimental periods were cross-cultivated with samples of donor and autologous stool filtrates obtained at the same time points (Figure 1b). Criteria for cell activation under conditions in vitro are the expression of the markers CD25 and CD127, corresponding to IL-2 and IL-7 receptors, on the cell surface. It has been shown that CD25^+^CD127^+^ are the helper T-cells with anti-inflammatory properties. In particular, cells producing IL-10 [20] and CD25^+^CD127^low^ cells are full-fledged regulatory T cells [21].

When isolating cells and stool filtrates, it was necessary to evaluate the initial indicators of the immune status of patients, as well as the presence of the studied cytokines in stool filtrates, to exclude artifacts in subsequent measurements ex vivo. No significant changes were found in the total content of CD4^+^ T lymphocytes in the blood of patients over the period after the therapy and compared with the group of healthy patients (Figure 2a). At the same time, analysis of the cell content phenotypically corresponding to regulatory T lymphocytes (CD3^+^CD4^+^CD25^+^CD127^low^) demonstrated a trend towards a decrease in the number of these cells in patients 3 months after therapy as compared with the state before therapy and compared to a group of healthy patients (Figure 2b). The content of Treg cells in the blood of patients before therapy did not significantly differ from that of healthy patients.

### 2.3. Influence of Sterile Fecal Filtrates from Donors on the Helper T Cell Population in UC Patients

At the first stage, the effect of sterile filtrates of donor stool on phenotypic and physiological changes in cells in vitro was assessed. The introduction of these filtrates into cell cultures obtained before and after therapy demonstrated time-dependent dynamics in relation to the presence of both activated T cells and their production of cytokines. Analysis of cytokine production (Figure 3a,b) indicated that sterile filtrates of donor stool were unable to induce production of the cytokine IL-4, which is the Th2 cell response marker, in the culture fluid of patients. The production of IL-10 was significantly increased in patients three months after therapy. One month after therapy, there was also a tendency for IL-10 to increase, but only in some patients. The sterile filtrates of donor stool were unable to induce IL-10 production in cells from patients before therapy.

Regarding the presence of CD4^+^ cells in culture, it indicated their increase in dynamics after therapy (Figure 3c). Analysis of the activation status of these cells in terms of expression of CD25 and CD127 markers showed a time-dependent increase in the content of activated lymphocytes with the CD25^+^CD127^+^ phenotype (Figure 3d). Normalizing their number to the total number of CD4^+^ cells reduced the level of the differences. Cells that were obtained before therapy responded to the addition of sterile filtrates of donor stool to a lesser extent than those that were obtained three months after therapy, which may indicate the formation of an antigen-specific immune response pattern. Cells of healthy volunteers responded to the addition of sterile filtrates of donor stool by increasing the content of CD4^+^ lymphocytes and CD4^+^CD25^+^CD127^+^ cells at a level corresponding to the cells of patients before therapy. Apparently, the sterile filtrate of donor stool, having a minimal antigenic and adjuvant load, was able to form an environment that supports the viability of immune cells; however, it was not sufficient to activate proliferation. No CD25^+^CD127^low^ cells were detected.

Analysis of the subtypes of activated cells characterizing the memory phenotype according to the expression of CD45RA, CD45RO, and CCR7 markers (Figure 4) demonstrated that the bulk of cells responding to the addition of the filtrate were TEMRA-cells (CD45RA^+^CD45RO^−^CCR7^−^), corresponding to maximally reactogenic cells, both when introducing sterile filtrate of donor stool into the cells of patients and into the culture of healthy volunteers. At the same time, the content of cells with a central memory phenotype increased in patients three months after therapy. This effect was not observed in cells from healthy volunteers. 

### 2.4. Autologous Sterile Fecal Filtrates Drive Changes in IL-4 and IL-10 Production 

The production of cytokines in the cultures of these cells is characterized by a noticeable dynamic in the production of IL-4 and IL-10 (Figure 5). Thus, in contrast to the sterile filtrate of donor stool, the Auto 0 and Auto 1 filtrates were able to induce the production of IL-4 in cells before therapy and were less pronounced in cells the first month after therapy. The Auto 3 filtrate was unable to significantly stimulate IL-4 production. At the same time, there was a decrease in sensitivity to stimulation of IL-4 production depending on the period of cell receipt.

On the contrary, IL-10 production resembled that during stimulation with sterile filtrates of donor stool, in which cells of the zero and first month did not respond to the addition of the filtrate, while cells of the third month significantly produced this cytokine. At the same time, there was a drop in the level of IL-10 production, depending on the time point at which the cells were obtained. 

The ratio of the levels of these cytokines in different groups shows a statistically significantly increased IL-10/IL-4 ratio in all groups, although its level decreased depending on the filtrate used. Unfortunately, the study design did not allow for investigating the production of cytokines by individual cells, which does not provide insight into the cellular source of these cytokines.

### 2.5. Autologous Sterile Fecal Filtrates Influence the Helper T Cell Population

To assess changes in the immunomodulatory properties of patient filtrates over time after therapy, the stool filtrates of patients were introduced into cultures of peripheral mononuclear cells. The experiment was organized in such a way that the filtrates obtained at each stage of this study were applied to cells of the same stage as well as to cells obtained in subsequent stages (Figure 1b). If SFMT therapy leads to the formation of a new microbiota with different antigenic properties, a change in the ability of cells to be activated in the presence of the filtrate of the subsequent period can be assumed.

The level of CD4^+^ lymphocytes in the culture fluid increased depending on the age of the cells, but not on the age of the filtrate that was added to the culture fluid. ANOVA analysis showed the presence of a statistically significant time factor for cells (F = 67.01, *p* value < 0.0001), while the time factor of filtrates introduced into the culture medium had practically no role (F = 1.43, *p* value = 0.27). This change can be traced both in a paired analysis of cell dynamics (Figure 6b) and in the total dynamics of the content of CD4^+^ lymphocytes (Figure 6a).

The change in the number of CD25^+^CD127^+^ cells (Figure 6c) was also practically independent of what filtrate was added to the cell culture (ANOVA, F = 1.2, *p* value = 0.31); however, when autologous filtrates of months one and three were added to the culture, a sharp gap was observed between the number of these cells among PBMCs of months one and three. At the same time, a pronounced scatter in the level of CD25^+^CD127^+^ was observed among cells of the third month, regardless of the stimulation option.

Analysis of cellular memory markers indicated the following patterns (Figure 7). Just as in the case of stimulation of cells with sterile filtrate of donor stool, stimulation with autologous filtrates was accompanied by a high content of TEMRA-cells, the level of which, however, fell among the cells of the third month. In this case, there was a clear increase in the number of these cells when stimulated with filtrates from the third month, compared with filtrates before therapy (Kruskal–Wallis, *p* value < 0.001). When stimulated with Auto 0 and Auto 1 filtrates, the level of TEMRA-cells in the cells of the third month was significantly lower than in the cells of the first and second months. A two-way ANOVA demonstrated the dependence of the severity of changes on which cells were introduced (F = 20.44, *p*-value < 0.001) and on what filtrates were used for stimulation (F = 6.49, *p*-value < 0.001).

The content of central and effector memory cells had a slight dependence on the type of filtrate introduced (corresponding ANOVA data: F = 4.89, *p* value = 0.063 and F = 4.03, *p* value = 0.009), but much more dependent on the time point when cells were obtained (F = 39.4 and F = 71.17, respectively, *p* value < 0.0001). It should be noted that in the cells of the third month, there was a pronounced dispersion in the level of content of these cell types.

This pattern resembles the differences in cell composition during stimulation with sterile filtrate of donor stool, in which, depending on the time point the cells were obtained, displacement of the TEMRA cell population was observed in favor of memory cell populations. In this case, sterile fecal filtrates (Auto 0 and Auto 1) shift to a greater extent.

## 3. Discussion

Sterile filtrates of the fecal microbiota from patients before and after therapy contain a factor that stimulates the formation of cells with the activation phenotype CD25^+^CD127^+^. Moreover, cells before therapy tend to produce IL-4, while cells after therapy and, in particular, cells of the third month, are more likely to produce IL-10. The production of IL-10 by these cells is most pronounced when stimulated with donor filtrate and patient filtrate before therapy. Sterile fecal filtrates from patients after three months do not cause a significant increase in IL-4 production and stimulate IL-10 production to the least extent. However, all sterile fecal filtrates were able to stimulate IL-10 in cells after the SFMT procedure.

Based on the data presented, it is possible to suggest a model of the influence of Sterile Fecal Microbiota Transplantation on the state of the patient’s immune system. Model 1 suggests that the sterile filtrate of donor stool introduces bacteriophages into the intestines of patients, which can change the ratio of the intestinal bacterial flora towards less pro-inflammatory bacteria. Model 2 states that the main effect of the sterile filtrate of donor stool is to modify microbiome metabolites, which in turn affect the proportion of bacteria in the patient’s gut. In any of these cases, the factor influencing the state of immune cells can have both antigenic and adjuvant properties. In the first case, the main therapeutic factor is the decrease in the number of bacteria causing persistent activation of pro-inflammatory signals. In the second case, there is a modification of the response of cells with the same immune specificity that was present in the intestines of patients.

It should also be taken into account that the exposure time of the donor material on the intestinal epithelium does not exceed 2–3 h, and therefore, the introduced factors either must be potent or their short-term effects must be accompanied by long-standing consequences. From this point of view, model 1 seems to be more acceptable. Provided that a suitable host bacterium is found and the corresponding bacteriophages express just moderate lytic properties, these phages are able to persist in the ecosystem for a long time, controlling the degree of growth of certain bacteria. The feasibility of such a model has been previously shown in a study of sterile microbiota transplantation in patients with clostridia infection [22] and is directed to a significant increase in bacteriophages of the Caudoviricetes class [22]. This effect is associated with a shift in the microbiota towards an increase in the representation of the Firmicutes phylum compared to Bacteroidetes [23].

While this model is capable of explaining the disease severity relief, it does not provide clues to understanding the helper T cell response skewness. Particularly, our data show that increased IL-10 production can be observed independently of what type of filtrate is introduced to the culture, indicating that the antigenic load is not as necessary as the period post-therapy. 

FMT has previously been shown to alter the ratio of helper T cells in favor of regulatory T cells and other IL-10-producing T cells [18]. It has been shown that microbiota transplantation in patients with inflammatory bowel diseases leads to an increase in the production of IL-10 and a decrease in IL-17 in the blood [19]. This effect was directly attributed to the colonization of healthy donor intestinal microbiota, accompanied by an increase in the production of a number of metabolites that promote the formation of regulatory T cells. However, other studies have shown that intestinal colonization in FMT is not a key factor in the beneficial effects of therapy, and some small molecules produced by the “healthy” microbiota are more important [12]. For example, it is now known that SCFA and bile acids create an environment to stimulate helper T cells towards regulatory phenotypes independently of antigen. On the other hand, the influence of low-molecular compounds is limited to a single use, which does not allow the formation of a long-term pharmacological effect. 

Perhaps the combination of antigen-specific and non-specific factors is the clue to solving the task of switching the helper T cell response pattern. However, more thorough studies are required to unravel this problem. Particularly, it should be uncovered which bacteriophages contribute to the microbiota shift and which of the representatives of the microbiota are capable of game-changing in the helper T cell phenotype definition. It is also to be disclosed whether the effect of SFMT is longstanding or if it has a limited duration. It is also likely that no single, defined formulation of SMFT content exists, and the therapy is about to be patient-specific. Nevertheless, SFMT shows promise as a therapy that influences the mechanisms rather than just the symptoms of UC. 

As for the limitations of this study, the small sample size of patients is the most important one due to the experimental nature of the procedure and strict inclusion criteria, with the exclusion of patients immediately before the planned procedure in case of deterioration or detection of systemic signs of an inflammatory response in clinical tests. Limitations also include the lack of a group of healthy volunteers who underwent a similar procedure. Subsequent studies are expected to take into account the above circumstances, including increasing the sample size of patients, including healthy volunteers in the study group, and also expanding the range of markers used to identify lymphocytic phenotypes. 

It is suggested that the mechanism of the immune response alteration is related to both innate and adaptive response modulation. The presence of small molecules, such as SCFA or components of bacterial cell walls, was introduced with donor stool filtrates. One interesting hypothesis is that the bacteriophage content of donor stool filtrate [24] modifies the microbial content of the recipient’s bowel. In that case, the decrease in the pro-inflammatory state of T cells may be attributed to the elimination of microbial strains serving as antigen sources for the given patient. However, it does not explain the sustained T-cell response 1 and 3 months after FMT. An alternative hypothesis is that bacteriophages may serve as the initial antigenic agent, priming T cells via stimulation of antigen-presenting cells. Following that, once bacteriophages are adopted, they may boost primed T-cells, sustaining the inflammatory condition. 

Overall, the obtained data proposes a potentially novel approach to the treatment of inflammatory conditions related to bowel diseases. The approach can be postulated as immune modulation by bowel immunization, as the observed changes in helper T-cell response skewness are progressive during the observational period after FMT, which indicates the antigen-specific character of the intervention.

## 4. Materials and Methods

### 4.1. Donors

All donors were young, healthy volunteers (20–25 years old), not suffering from chronic diseases, free of infections, and have not had hospitalization incidents for at least the last two months. All potential donors were examined with biochemical and microbiological assays, including general and biochemical blood tests, as well as blood ELISA for the presence of Giardia, Toxocara, Opisthorchid, Ascaris, and Trichynella. Additionally, donors were examined for the infections *T. pallidum* (“Vector-Best” LLC, Novosibirsk, Russia), HIV 1 and HIV 2 (“Vector-Best” LLC, Novosibirsk, Russia), and hepatitis B and C viruses (“Vector-Best” LLC, Novosibirsk, Russia). All donor fecal samples were assessed for the presence of *Salmonella* spp., *Shigella* spp., enteroinvasive *Escherichia coli*, and *Cryptosporidium* spp. (by routine microbiological methods), as well as helminths and their eggs by microscopy. *C. difficile* toxins A and B were detected using an immunochromatographic rapid test (Toxin A + B (Clostridium difficile) DUO, Vedalab, Alençon, France). In addition, DNA of *Shigella* spp., enteroinvasive *Escherichia coli*, *Salmonella* spp., and *Campylobacter* was tested by PCR with hybridization-fluorescence detection of amplification products (Amplisence OKI Bacto-Screen-FL, Moscow, Russia); nucleic acids of rotaviruses A, noroviruses I and II, and adenoviruses F were screened by the same method (Amplisence OKI Viro-Screen-FL, Moscow, Russia). In addition, donors were screened for *H. pylori*.

This study was approved by the Local Ethics Committee of the Autonomous Non-Commercial Organization “Center of New Medical Technologies in Akademgorodok” (Protocol #2, date of approval: 12 January 2019). Informed written consents were obtained from donors and patients before enrollment as well as during the follow-up period.

### 4.2. Patients 

Adult patients of any gender with UC of mild severity level (UCEIS score 2–4) were recruited in the Center for New Medical Technologies SB RAS during the period from December 2021 to January 2023. The inclusion criteria were UC score of 2–6, confirmed with clinical assessment and colonoscopy, and an ESR level less than 30 mm/hour. All patients were screened for *Clostridioides difficile* or *Helicobacter pylori*. Then, the patient was referred for an electrocardiogram and a consultation with a therapist and a gastroenterologist. (The anesthesiologist examined the patient before the colonoscopy.) In addition, stool samples from each involved patient before treatment were examined for all pathogens listed in Section 2.1. Exclusion criteria: age (under 18 and over 79); pregnancy; confirmed diagnosis of irritable bowel syndrome or Crohn disease; the presence of *C. difficile* or *H. pylori*. In addition, STMT was not performed if the patient had an increased stool frequency, ESR rate, and blood signatures of exacerbation of the inflammatory response (increased levels of CRP and WBC) on the day of the procedure or early during the curation period. All patients were followed up during the period of 3 months after the procedure. The blood parameters and colonoscopy were performed. 

### 4.3. Preparation of Sterile Fecal Filtrate 

Donor stool samples were collected in a clean, specifically prepared environment. Part of the stool samples were used for clinical laboratory assessment; the other part was divided into samples of equal weight (over 100 g) and frozen at −84 °C. 

Donor samples were used for the preparation of the sterile filtrate the day before the procedure of sterile fecal transplantation. The procedures were conducted under the standard operational procedures developed locally in the Center for Novel Medicine Technologies in clean, aseptic conditions. Briefly, the thawed portion of donor stool was placed in 300 mL of sterile clinical-grade solution of 0.9% NaCl, blended using a tissue shredder, and sequentially filtered through sterile filter cascades of 0.45 and 0.22 micrometer filter cassettes into the GMP-grade sterile disposable flask. Filtrates were stored at +4 °C overnight. Two hours before the procedure of Sterile Fecal Microbiota Transplantation, the material was placed in an aseptically processed box to equilibrate the temperature. The filtrates were introduced into the colons of patients during the fibrocolonoscopy procedure.

### 4.4. Patient’s Sample Collection

Blood and stool samples were collected before the procedure of Sterile Fecal microbiota transplantation and one and three months after the procedure. For this study, donors and patients were given venous blood early in the morning. Blood was transported to this research laboratory in blood collection tubes with an anticoagulant (K3 EDTA). Stool samples were frozen at −20 °C until the analysis.

### 4.5. PBMC and Plasma Preparation from Patient Venous Blood

Cell fractionation was performed within 2 h after sampling. At the first stage, it was centrifuged at 300× *g* for 10 min, and a fraction of the crystals was taken to a depth of 1.5–2 mL. Next, the rest of the sample was diluted twice in calcium-free phosphate-buffered saline and transferred to 15 mL conical tubes pre-filled with Ficoll Histopaque-1077 solution at a ratio of blood:Ficoll = 3:1. Next, the solutions were centrifuged at 400× *g* for 30 minutes in a centrifuge with a rotary bucket type at a reduced speed of acceleration and deceleration. After centrifugation, the upper fraction containing diluted plasma and platelets was removed, and then the interphase containing peripheral mononuclear cells was collected. This pool was washed three times in phosphate-buffered saline, after which the cells were counted. Some of the cells were immediately stained with marker antibodies for immunophenotyping.

### 4.6. PBMC Freezing and Storage

The cells were frozen in a solution containing fetal bovine serum and DMSO (in a ratio of 9:1) with a density of 5 million/mL. Freezing was carried out in two stages: (1) in containers with isopropanol up to −70 °C; and (2) transfer of frozen test tubes to liquid nitrogen vapor. The plasma samples were frozen at −70 °C.

### 4.7. Cultivation of PBMC 

Cultivation of isolated peripheral mononuclear cells was carried out in RPMI-1640 medium containing inactivated calf serum and glutamine (4 mM). On the day of the study, the cells collected at different periods of the study were thawed, washed from the freezing liquid, and placed in a prepared culture medium containing GM-CSF (at a final concentration of 40 ng/mL), which is necessary for the activation of monocytes and the presentation of antigens. The cells were seeded into the wells of a 96-well plate in the amount of 200 thousand cells per well in 100 μL of culture medium. Cells were supplemented with samples of sterile-filtered stool from donors or autologous. Cultivation was carried out for 4 days. During cultivation, the formation of cell clusters was observed, and their number and size were photographed. At the end of the cultivation period, the culture liquid was collected for enzyme immunoassay. Cells were stained with surface antibodies to analyze the phenotypes involved in the activation process.

### 4.8. FACS Analysis of Fresh and Cultured PBMC 

To determine lymphocyte subpopulations, peripheral mononuclear cells were stained with antibodies to surface markers (BD Biosciences, Franklin Lakes, NJ, USA). The following antibodies were used for this research panel: CD3-(FITC), CD4-(BV510), CD25-(PE), CD197-(BV421), CD45RA-(PerCP), and CD45RO-(APC-Cy7). This panel aims to search for the presence of regulatory T cells as well as their characterization in terms of memory phenotype and effector properties.

Staining was carried out in a phosphate-buffered saline solution containing 0.5% bovine albumin for 1 h at +4 °C. The cells were then fixed in 4% formalin for 15 min. Formalin was inactivated with a 0.15 M solution of glycine. Analysis was performed on a Novocyte 3000 flow cytometer (ACEA Biosciences Inc., San Diego, CA, USA).

### 4.9. Study of Cytokines in Blood and Sterile Stool Samples 

The study of cytokines in blood plasma was carried out by solid-phase ELISA using Vector-Best kits according to the instructions. The day before the study, the plasma was thawed at +4 °C. For the analysis of cytokines in sterile stool samples, the volume of the sample added to the ELISA plate was normalized to the weight of the sample from which the filtrate was obtained.

### 4.10. Data Analysis and Statistical Inference 

Flow Cytometry data analysis were performed using NovoExpress v.1.6.1 cytometry software. Data analysis and statistical inference were managed using GraphPadPrism v.8 and Python 3.7. For time-dependent paired analysis, a paired *t*-test with Tukey post-hoc was applied. For the comparison of stool filtrates, Kruskall–Wallis and Dunn pos-hoc were applied.

## 5. Conclusions

A pilot study of the T cell response to transplantation of sterile feces in ulcerative colitis patients is reported. This study was intended to evaluate the patterns of T cell responses that could potentially correspond to positive outcomes of the therapy. The hypothesis of this study was that the transfer of antigenic or adjuvant-like substances (but not microbial cells) with sterile fecal filtrates may lead to a transition of the state of T-cells from pro-inflammatory to anti-inflammatory or anergic. Our findings provide primary support for the hypothesis that peripheral T-cells change the type of response during the SFMT procedure. Thus, we observed a shift from the predominant production of IL-4 to the production of IL-10 when peripheral T cells were stimulated with sterile filtrate under ex vivo conditions. It was shown that cells with the CD3^+^CD4^+^CD25^+^CD127^+^ phenotype primarily survived during the stimulation ex vivo, with the assessed memory phenotype being mainly TEMRA cells and effector memory cells. Even though the data requires a more thorough study, particularly the determination of the components of sterile stool filtrates, as far as we know, this is the first observation demonstrating the direct influence of components of sterile stool filtrate on the state of T cells in patients with ulcerative colitis. The observations can potentially serve as a basis for the development of a novel type of anti-inflammatory therapy not only in UC patients but also in patients suffering from other inflammatory bowel diseases.

## Figures and Tables

**Figure 1 ijms-25-01886-f001:**
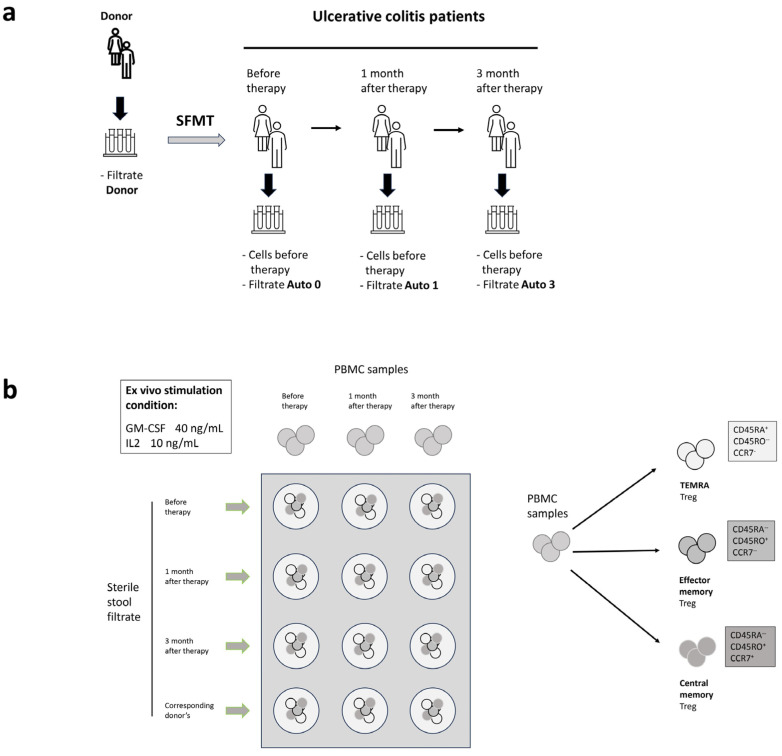
Experimental design. (**a**) Sterile filtrates of donor stools were obtained and collected at −84 °C. A day before the SFMT procedure, donor samples were thawed and processed under clean-room conditions. (**b**) Patient’s samples (blood and stool) were collected right before the SMFT and in the follow-up periods of 1 and 3 months.

**Figure 2 ijms-25-01886-f002:**
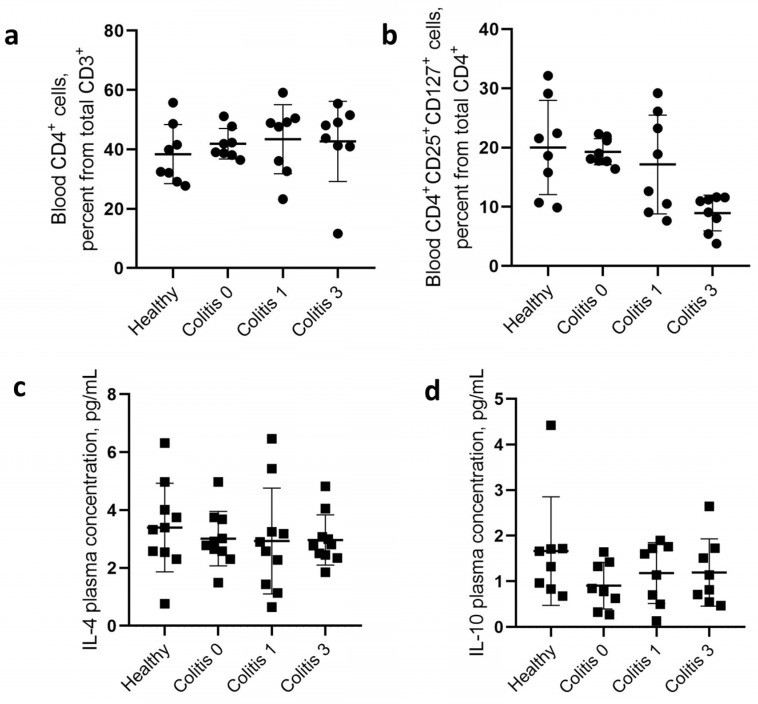
Patient and healthy control group CD4^+^ cells and cytokine levels. Plots description: (**a**) Overall CD4^+^ T-cell content among all CD3^+^ cells. (**b**) presence of T-helper cells with activation phenotype CD25^+^CD127^+^; (**c**) IL-4 plasma concentration plot; (**d**) IL-10 plasma concentration plot. Designations: Healthy—healthy control volunteers; Colitis 0—Blood samples from patients before the SFMT; Colitis 1 and Colitis 3—Blood samples from patients after 1 and 3 months post SFMT, respectively.

**Figure 3 ijms-25-01886-f003:**
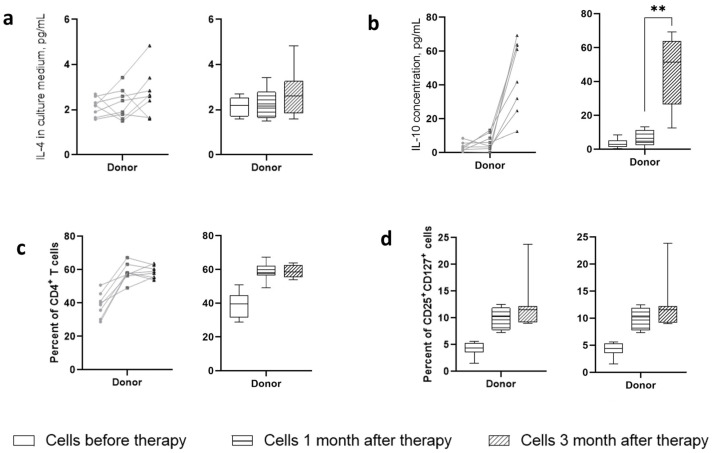
Influence of sterile fecal filtrates from donors on patients’ cells. Cells were stimulated in presence of GM-CSF and IL-2 for two days. Each metric is shown in both pairwise (dotted lines) and median-wise plots to illustrate the intra-individual dynamics of the parameter and the global dynamics over the study group. (**a**) IL-4 production in culture medium; (**b**) IL-10 production in culture medium; (**c**) presence of CD4^+^ T cells among the population survived after two days of ex vivo activation; (**d**) level of CD4^+^CD25^+^CD127^+^ cells capable of producing effector cytokines. **—designate significance level *p* < 0.001.

**Figure 4 ijms-25-01886-f004:**
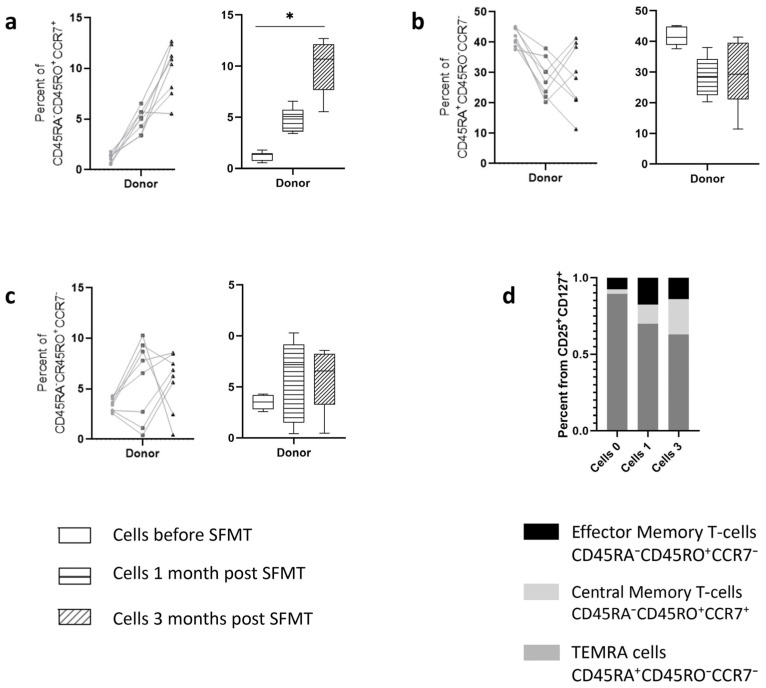
Memory phenotypes of CD25^+^CD127^+^ T cells in ex vivo cultured conditions under stimulation with donor-derived stool filtrate. Each metric is shown in both pairwise (dotted lines) and median-wise plots to illustrate the intra-individual dynamics of the parameter as well as the global dynamics over the study group. Plots description: (**a**) percent of activated T cells with phenotype of Central Memory cells (CD45RA^−^CD45RO^+^CCR7^+^); (**b**) Percent of cells with phenotype of TEMRA-Cells’ + 5; (**c**) Subset of cells with phenotype of Effector Memory cells. (**d**) Relative content of memory cell phenotypes in culture medium. Effector memory cells percent of *—designate significance level *p* < 0.01.

**Figure 5 ijms-25-01886-f005:**
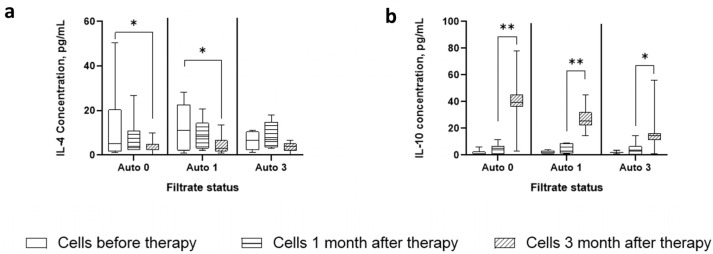
Dynamics of cytokine production in cells of different time points stimulated with autologous filtrates of different time points. (**a**) IL-4 concentration plots; (**b**) IL-10 concentration plots. * designates *p*-value < 0.01, ** *p*-value < 0.001. Auto 0, Auto 1, and Auto 3 stand for Autologous stool samples taken right before therapy, 1 month after therapy, and 3 months after therapy.

**Figure 6 ijms-25-01886-f006:**
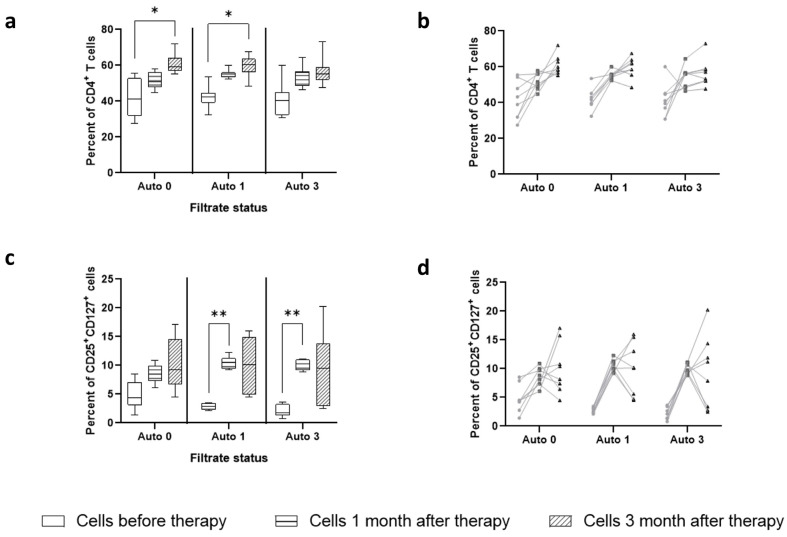
Dynamics of patient’s cells stimulated with autologous filtrates of different time points. Each metric is shown in both pairwise (dotted lines) and median-wise plots to illustrate the intra-individual dynamics of the parameter as well as the global dynamics over the study group. (**a**) Content of CD4^+^ T-cells in cell culture population depends on time the cells were obtained or the sterile stool filtrate applied to the cell culture. (**b**) The same data presented pairwise to demonstrate patient-specific dynamics. (**c**) Activation of CD4 T-cells based on CD25 and CD127 surface expression profiles. (**d**) The same data presented pairwise to demonstrate patient-specific dynamics. ** indicates significance level *p* < 0.001, * indicates significance level *p* < 0.01.

**Figure 7 ijms-25-01886-f007:**
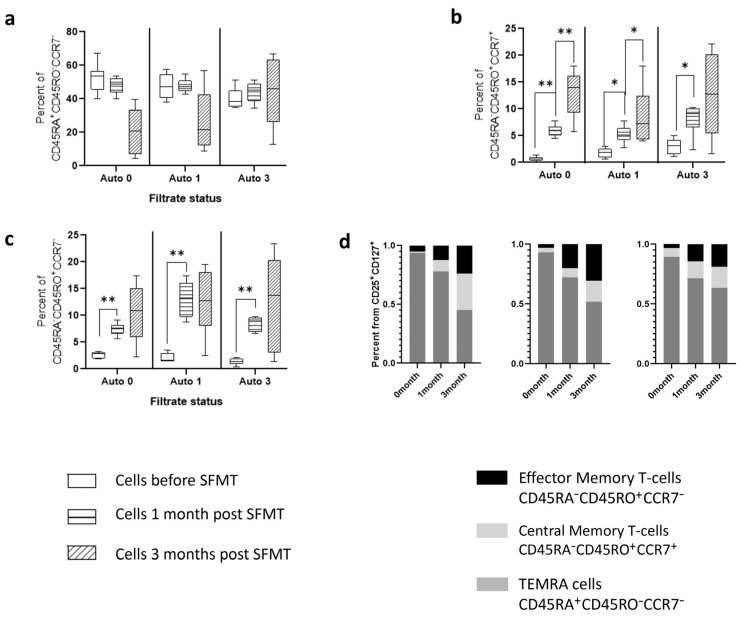
Memory phenotypes of CD25^+^CD127^+^ T cells in ex vivo cultured conditions under stimulation with donor-derived stool filtrate. Plots description: (**a**) percent of activated T cells with phenotype of Central Memory cells (CD45RA^−^CD45RO^+^CCR7^+^); (**b**) Percent of cells with phenotype of TEMRA-Cells’ + 5; (**c**) Subset of cells with phenotype of Effector Memory cells; (**d**) Relative content of memory cell phenotypes in culture medium. Auto 0, Auto 1, and Auto 3 correspond to patient’s sterile stool filtrate derived before therapy, and 1, and 3 months in the following period. **—designate significance level *p* < 0.001, *—designate significance level *p* < 0.01.

**Table 1 ijms-25-01886-t001:** The condition of patients with UC that were included in this study.

Patient	Sex	Age	Mayo Score	Fecal Calprotectin, μg/g	C-Reactive Protein, mg/L	Number of Defecations per Day	Presence of Blood in the Stool	Blood Hemoglobin, g/L
1	M	32	4	600	0.89	2–4	+	125
2	F	30	2	180	1.59	1	-	117
3	F	30	5	600	1.67	2–4	Traces	130
4	M	34	4	600	6.45	4	Traces	128
5	F	22	2	46	0.68	4	Traces	121
6	M	22	6	1056	5.93	20	+	133
7	F	20	6	901	3.27	10	+	92
8	F	51	6	563	4	5	+	117

## Data Availability

The data presented in this study are available on request from the corresponding authors.

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
