# Peer review of "Sterile Fecal Microbiota Transplantation Boosts Anti-Inflammatory T-Cell Response in Ulcerative Colitis Patients"

_ijms, 2024, doi:10.3390/ijms25031886_

Round 1

Reviewer 1 Report

Comments and Suggestions for Authors

Thank you for submitting your manuscript. The findings of your study provide promising insights into new therapeutic approaches for ulcerative colitis (UC). Your research indicates that Sterile Fecal Microbiota Transplantation (SFMT) can induce an anti-inflammatory shift in T-helper cell responses in individuals with UC, with this effect persisting for at least three months post-procedure. Furthermore, the study highlights the potential of sterile stool filtrates in influencing anti-inflammatory immune mechanisms, possibly paving the way for novel treatments for UC. Below are some suggestions for your article:

Line 42: You mention a hypothesis regarding immune system activity under inflammatory conditions. Please provide a sound experimental basis for this hypothesis and present it directly in the text to enhance the scientific rigor of your argument.

Line 71: In discussing the primary disadvantages of FMT, consider including references to ethical concerns to provide a comprehensive evaluation of FMT's application.

Line 110 and Line 215: Please elucidate the rationale behind co-culturing PBMCs with sterile stool filtrate, aiming to simulate a specific in vivo mechanism. This clarification will help in making your experimental approach more transparent.

Line 337 and Line 352: Regarding the description of patient enrollment, it is advisable to include a table for patient selection and screening, presenting detailed and clear information on the clinical state assessments of enrolled patients. Please articulate the theoretical justification for excluding patients with severe UC, as including patients with varying degrees of UC severity could demonstrate the broader applicability of SFMT.

Discussion Section: Given the publication of many high-impact FMT clinical studies, it would be beneficial to supplement your discussion with how your experimental approach or findings may offer potential improvements to the forefront of clinical research. Additionally, considering the therapeutic prospects for various inflammatory diseases of the digestive system, discussing in your paper how your experiment might provide insights for the treatment of other inflammatory conditions, such as acute pancreatitis, could attract attention from researchers in other fields.

I look forward to your revisions in response to these suggestions, aiming to enhance the quality and impact of your paper.

Yours sincerely.

Author Response

We would like to thank the Reviewer for good advices and hope that our corrections improve the manuscript

Comments and Suggestions for Authors

Thank you for submitting your manuscript. The findings of your study provide promising insights into new therapeutic approaches for ulcerative colitis (UC). Your research indicates that Sterile Fecal Microbiota Transplantation (SFMT) can induce an anti-inflammatory shift in T-helper cell responses in individuals with UC, with this effect persisting for at least three months post-procedure. Furthermore, the study highlights the potential of sterile stool filtrates in influencing anti-inflammatory immune mechanisms, possibly paving the way for novel treatments for UC. Below are some suggestions for your article:

Line 42: You mention a hypothesis regarding immune system activity under inflammatory conditions. Please provide a sound experimental basis for this hypothesis and present it directly in the text to enhance the scientific rigor of your argument.

Response: We have included the reference [2] to the statement.

https://www.tandfonline.com/doi/full/10.1080/00365521.2018.1447597

Line 71: In discussing the primary disadvantages of FMT, consider including references to ethical concerns to provide a comprehensive evaluation of FMT's application.

  • Response: We have provided the reference in the main text to the review concerning the advantages and disadvantages of FMT. [13]

Line 110 and Line 215: Please elucidate the rationale behind co-culturing PBMCs with sterile stool filtrate, aiming to simulate a specific in vivo mechanism. This clarification will help in making your experimental approach more transparent.

  • Response: The primary idea behind the given experimental design implies that metabolic and non-vital antigenic load of sterile stool filtrates may bear immune-stimulating or immune-modulating properties. The basic hypothesis is provided in the section 2.1

Line 337 and Line 352: Regarding the description of patient enrollment, it is advisable to include a table for patient selection and screening, presenting detailed and clear information on the clinical state assessments of enrolled patients. Please articulate the theoretical justification for excluding patients with severe UC, as including patients with varying degrees of UC severity could demonstrate the broader applicability of SFMT.

Response: We add a special Table 1 describing the condition of patients before sterile FMT. This table contains information on the clinical state assessments of enrolled patients. Patients with severe UC were excluded from the study based on several considerations: 1) In order to unify the experimental group; 2) Severe UC is typically provided with severe immune suppressive therapy that may inadequately interfere with the study aims; 3) Severe UC patients are of potentially greater risk of exacerbation of inflammation due to increased immune system response, requiring specialized medical support and increased observation time that was inacceptable within the current study. This theoretical justification is added to the Section 2.1. Patients.

Discussion Section: Given the publication of many high-impact FMT clinical studies, it would be beneficial to supplement your discussion with how your experimental approach or findings may offer potential improvements to the forefront of clinical research. Additionally, considering the therapeutic prospects for various inflammatory diseases of the digestive system, discussing in your paper how your experiment might provide insights for the treatment of other inflammatory conditions, such as acute pancreatitis, could attract attention from researchers in other fields.

- Response: We supplement the Discussion section with additional speculation.

I look forward to your revisions in response to these suggestions, aiming to enhance the quality and impact of your paper.

Yours sincerely.

Reviewer 2 Report

Comments and Suggestions for Authors

ijms-2786906 Review December 2023: Sterile Fecal Microbiota Transplantation Boosts Anti- Inflammatory T-Cell Response in Ulcerative Colitis Patients.

The article discusses an interesting topic such as fecal transplants to treat inflammatory bowel diseases. In this case they emphasize the use of sterile fecal microbiota transplants, and how these can help the inflammatory process. The article is interesting and well structured. I have found some typographical errors and I think that including the conclusion in its own section can help understand the text. Homogenizing the presentation of results and taking care of the writing of the text are necessary.

Key words

Key words shouldn’t appear in title.

Material and methods

Line 372: do-nors, should be donors

Line 414: Possible?

Line : per-formed, should be performed

Results

Line 101,119, 138, 217, 221,346: Use impersonal language

Line 104, 139,272: in vitro, ex vivo… should be in italic

Conclusions

Although it is not mandatory, I think that in this case, including a conclusion section can help follow the text. The authors write their conclusions at the end of the discussion, they could differentiate it in their own section.

Figures and tables

Figure 3c should present individual data as the others.

Line 185: missing d)

Recommending to accept the work for publication

Author Response

We would like to thank the Reviewer for good advices and hope that our corrections improve the manuscript

Comments and Suggestions for Authors

ijms-2786906 Review December 2023: Sterile Fecal Microbiota Transplantation Boosts Anti- Inflammatory T-Cell Response in Ulcerative Colitis Patients.

The article discusses an interesting topic such as fecal transplants to treat inflammatory bowel diseases. In this case they emphasize the use of sterile fecal microbiota transplants, and how these can help the inflammatory process. The article is interesting and well structured. I have found some typographical errors and I think that including the conclusion in its own section can help understand the text. Homogenizing the presentation of results and taking care of the writing of the text are necessary.

Key words

Key words shouldn’t appear in title.

  • Response: The number of key words is reduced

Material and methods

Line 372: do-nors, should be donors

Line 414: Possible?

Line : per-formed, should be performed

  • All these typos have been corrected.

Results

Line 101,119, 138, 217, 221,346: Use impersonal language

Line 104, 139,272: in vitro, ex vivo… should be in italic

- Response: Impersonal language is now used in these sentences. Italic is used too.

Conclusions

Although it is not mandatory, I think that in this case, including a conclusion section can help follow the text. The authors write their conclusions at the end of the discussion, they could differentiate it in their own section.

  • Response: The conclusions section is attached

Figures and tables

Figure 3c should present individual data as the others.

Line 185: missing d)

  • Response: Individual data are present. The typo has been corrected.

Reviewer 3 Report

Comments and Suggestions for Authors

I congratulate the authors, the work is well structured and scientifically valid. I find it very interesting given that  ulcerative colitis diffusion. I have just a few considerations.

Minor questions:

1.       Throughout the text the contracted name of the various cytokines has been reported without the hyphen. (IL4, IL10, IL2 etc), please correct with IL-4, IL-10, IL-2 etc.

2.       In the introduction section (lines 33 - 37) the type 4 hypersensitivity response which appears to be fundamental in ulcerative colitis was mentioned. Briefly describe the type 4 hypersensitivity response. 2 lines of comment are enough, perhaps giving the example of DTH or contact hypersensitivity. Just two lines mentioning the cells involved are enough.

3.       In relation to the comments in lines 142 - 148, plots relating to the production of IL-4 and IL-10 are shown in figures 3a and 3b. In both cases the words "IL-10 concentration pg/ml" or "IL-4 concentration pg/ml" are shown on the Y axis. So you don't understand the difference between the right and left plot. I assume that the box plot indicates the median concentration with the interquartile range in the 3 groups (divided at time zero, 1 month and 3 months after treatment), while the broken line graph is evidently the trend for paired data. The authors should be more precise and adjust the wording on the axes of the graphs.

4.       As above, plots relating to the levels of CD4+CD25+CD127+ are shown in figure 3d. The writing on the Y axis in both graphs (left and right) is the same. What do the authors want to show? The percentage of total CD4 in the box plot and the trend for paired data in the broken line graph? This must be clarified by changing the wording on the axes of the graphs, because this can lead to confusion.

5.       With reference to the comments between lines 172 - 179, As above, in figure 4 plots are shown relating to the levels of CD4/CD45RA/CD45RO/CCR7. The writing on the Y axis in both graphs (left and right) is the same. What do the authors want to show? This must be clarified by changing the wording on the axes of the graphs, because this can lead to confusion. Furthermore, in figure 4a in the pair of graphs the wording is shown on the Y axis, in 4b and 4c, but the legend is missing on the Y axis of the second graph of each copy, please add it and align it with figure 4a. Where paired data is shown it would be appropriate to indicate it on the axis of the graph to make it better understood.

6.       In reference to what was commented in lines 228 - 236, what do graphs 6b and 6d indicate in figure 6? From the legend it can be seen that data are indicated by pairing but the wording on the graph axis should be corrected by also indicating it in the graph axis legend.

7.       In the materials and methods section, the authors describe (lines 366 - 379) the analyzes to which the subjects used in the present study were subjected. Has the calprotectin measurement on the patients' faeces (before and after treatment) been carried out? Since it is a useful marker of inflammatory activity and is used to follow the progress of inflammatory bowel disease, it would be interesting to see its progress. If this data is available it should be added to enrich the work.

8.       In the materials and methods section (lines 431 - 441) the authors describe the analyzes conducted on PBMCs and the related labeling for T lymphocyte typing. I was wondering if the authors had tried to carry out intracellular labeling on the stimulated PBMCs (after addition of monesin and brefeldin) to verify the production of cytokines such as IL-17A, IFN gamma, IL-4, IL-10, TNF alpha and IL-2. Obviously on CD4, CD40L, CD137. This is to have an indication of the activation status. If these data had been collected they should be added to enrich the work, because it would be very interesting, especially 1 month after treatment (compared to T0).

Author Response

We are grateful to the Reviewer as the obtained comments helped us to improve our manuscript.

Comments and Suggestions for Authors

I congratulate the authors, the work is well structured and scientifically valid. I find it very interesting given that  ulcerative colitis diffusion. I have just a few considerations.

Minor questions:

  1. Throughout the text the contracted name of the various cytokines has been reported without the hyphen. (IL4, IL10, IL2 etc), please correct with IL-4, IL-10, IL-2 etc.

- Corrected

  1. In the introduction section (lines 33 - 37) the type 4 hypersensitivity response which appears to be fundamental in ulcerative colitis was mentioned. Briefly describe the type 4 hypersensitivity response. 2 lines of comment are enough, perhaps giving the example of DTH or contact hypersensitivity. Just two lines mentioning the cells involved are enough.

                - The description is attached (lines 35-37)

  1. In relation to the comments in lines 142 - 148, plots relating to the production of IL-4 and IL-10 are shown in figures 3a and 3b. In both cases the words "IL-10 concentration pg/ml" or "IL-4 concentration pg/ml" are shown on the Y axis. So you don't understand the difference between the right and left plot. I assume that the box plot indicates the median concentration with the interquartile range in the 3 groups (divided at time zero, 1 month and 3 months after treatment), while the broken line graph is evidently the trend for paired data. The authors should be more precise and adjust the wording on the axes of the graphs.

                - Descriptions is clarified

  1. As above, plots relating to the levels of CD4+CD25+CD127+ are shown in figure 3d. The writing on the Y axis in both graphs (left and right) is the same. What do the authors want to show? The percentage of total CD4 in the box plot and the trend for paired data in the broken line graph? This must be clarified by changing the wording on the axes of the graphs, because this can lead to confusion.

- Plot is reshaped

  1. With reference to the comments between lines 172 - 179, As above, in figure 4 plots are shown relating to the levels of CD4/CD45RA/CD45RO/CCR7. The writing on the Y axis in both graphs (left and right) is the same. What do the authors want to show? This must be clarified by changing the wording on the axes of the graphs, because this can lead to confusion. Furthermore, in figure 4a in the pair of graphs the wording is shown on the Y axis, in 4b and 4c, but the legend is missing on the Y axis of the second graph of each copy, please add it and align it with figure 4a. Where paired data is shown it would be appropriate to indicate it on the axis of the graph to make it better understood.

- Descriptions is clarified

  1. In reference to what was commented in lines 228 - 236, what do graphs 6b and 6d indicate in figure 6? From the legend it can be seen that data are indicated by pairing but the wording on the graph axis should be corrected by also indicating it in the graph axis legend.
  2. In the materials and methods section, the authors describe (lines 366 - 379) the analyzes to which the subjects used in the present study were subjected. Has the calprotectin measurement on the patients' faeces (before and after treatment) been carried out? Since it is a useful marker of inflammatory activity and is used to follow the progress of inflammatory bowel disease, it would be interesting to see its progress. If this data is available it should be added to enrich the work.

-              Response: We add Table 1 describing the condition of patients before sterile FMT. This table contains information on the clinical state assessments of enrolled patients, including the fecal calprotectin concentration.

  1. In the materials and methods section (lines 431 - 441) the authors describe the analyzes conducted on PBMCs and the related labeling for T lymphocyte typing. I was wondering if the authors had tried to carry out intracellular labeling on the stimulated PBMCs (after addition of monesin and brefeldin) to verify the production of cytokines such as IL-17A, IFN gamma, IL-4, IL-10, TNF alpha and IL-2. Obviously on CD4, CD40L, CD137. This is to have an indication of the activation status. If these data had been collected they should be added to enrich the work, because it would be very interesting, especially 1 month after treatment (compared to T0).

                - The initial study design did not imply measurement of intracellular cytokines as well as the spectrum of activation markers. That study was proposed as observational with respect to the immune system activity to clarify the types of cells implicated in the response. We intentionally used CD25 and CD127 as markers potentially indicating expansion of regulatory T cells, serving simultaneously as an activation markers for T cells. During the next patient enrollment we are going to specifically focus on the comparison of IL-4, IL-17 and IL-10 producing cells activation status to evaluate potential antigenic or non-antigenic (such as PAMPs) source for the activation or suppression of these types of cells.
